# Isolation, Partial Characterization and Application of Bacteriophages in Eradicating Biofilm Formation by *Bacillus cereus* on Stainless Steel Surfaces in Food Processing Facilities

**DOI:** 10.3390/pathogens11080872

**Published:** 2022-08-02

**Authors:** Maroua Gdoura-Ben Amor, Antoine Culot, Clarisse Techer, Mousa AlReshidi, Mohd Adnan, Sophie Jan, Florence Baron, Noël Grosset, Mejdi Snoussi, Radhouane Gdoura, Michel Gautier

**Affiliations:** 1Laboratory Research of Toxicology-Microbiology Environmental and Health LR17ES06, Sciences Faculty of Sfax, University of Sfax, Sfax 3000, Tunisia; gdoura.radhouane@gmail.com; 2Equipe Microbiologie, Agrocampus Ouest, INRA, UMR1253 Science et Technologie du Lait et de l’Œuf, 35042 Rennes, France; antoine.culot@mixscience.eu (A.C.); sophie.jan@agrocampus-ouest.fr (S.J.); florence.baron@agrocampus-ouest.fr (F.B.); noel.grosset@agrocampus-ouest.fr (N.G.); michel.gautier@agrocampus-ouest.fr (M.G.); 3Mixscience, Rue des Courtillons, ZAC Cissé Blossac, 35712 Bruz, France; techerclarisse@yahoo.fr; 4Department of Biology, College of Science, University of Hail, Ha’il P.O. Box 2440, Saudi Arabia; drmohdadnan@gmail.com (M.A.); snmejdi@yahoo.fr (M.S.); 5Laboratory of Genetics, Biodiversity and Valorisation of Bioressources, High Institute of Biotechnology, University of Monastir, Monastir 5000, Tunisia

**Keywords:** *B. cereus* group, biofilm, stainless steel, bacteriophage, sewage, characterization, eradication, inhibition

## Abstract

The *Bacillus cereus* (*B. cereus*) group is a widespread foodborne pathogen with a persistent ability to form biofilm, and with inherent resistance to traditional treatment in the food industry. Bacteriophages are a promising biocontrol agent that could be applied to prevent or eliminate biofilms formation. We have described, in this study, the isolation from sewage samples and preliminary characterization of bacteriophages that are active against the *B. cereus* group. The effectiveness of phage treatment for reducing *B. cereus* attachment and biofilms on stainless steel surfaces has been also assessed using three incubation periods at different titrations of each phage. Out of 62 phages isolated, seven showed broad-spectrum lytic action against 174 *B. cereus* isolates. All selected phages appeared to be of the *Siphoviridae* family. SDS-PAGE proved that two phages have a similar profile, while the remainder are distinct. All isolated phages have the same restriction pattern, with an estimated genome size of around 37 kb. The isolated bacteriophages have been shown to be effective in preventing biofilm formation. Reductions of up to 1.5 log_10_ UFC/cm^2^ have been achieved, compared to the untreated biofilms. Curative treatment reduced the bacterial density by 0.5 log_10_ UFC/cm^2^. These results support the prospect of using these phages as a potential alternative strategy for controlling biofilms in food systems.

## 1. Introduction

*B. cereus* group bacteria are widely distributed in the environment. Several species from the *B. cereus* group are associated with food poisoning cases by producing two different kinds of toxins: an emetic one and a diarrheal one [1]. *B. cereus* can be found in biofilm form in the food industry. Biofilms consist of complex communities of surface attached bacteria, surrounded by a polymeric extracellular material that typically consists of extracellular polysaccharides, proteins and sometimes eDNA [2]. The extracellular matrix of a biofilm protects the involved bacteria from environmental damage, antimicrobial agents, and host immune defenses [3]. Therefore, biofilms are extremely difficult to remove, exhibiting resistance to cleaning and disinfecting processes, as is the case of *B. cereus* biofilms, which exhibit resistance to the action of quaternary ammonium compound and sodium hypochlorite [4]. The resistance of *B. cereus* to chemical treatments is due to the bacterium’s ability to form spores and produce toxins that, in turn, can be harmful to humans and the food industry [5,6]. Thus, the negative impact of biofilm formation on food safety and economic losses in food processing [7,8] have enhanced the development of different new approaches in order to control and/or eradicate biofilm formation, such as the use of enzymes, bacteriophages, microbial interactions, and metabolic molecules [9]. Recently, new research on the use of bacteriophages as potential agents for biofilm eradication has increased. Indeed, several reports on the interactions between phages and biofilms have been published for a wide range of bacteria, such as: *Escherichia coli* [10], *Enterococcus faecalis* [11], *Listeria monocytogenes* [12,13], *Pseudomonas fluorescens* [14], *Staphylococcus aureus* [15], *Salmonella* [16,17,18,19], and *Streptococcus mutans* [20]. 

Bacteriophages, more commonly known as phages, are obligate parasites of bacteria. They are ubiquitous in nature, and they have been isolated from different environments including soil, water, and several food products such as meat, dairy, and vegetable products [21,22,23,24,25,26]. Several bacteriophage infecting *B. cereus* group bacteria have been isolated from soils, muds, food wastes and fermented foods [27,28,29,30]. Although there have been, as cited above, several studies published investigating the effect of phages against biofilms, no study has proposed phages as alternative biocontrol agents against *B. cereus* biofilm. Therefore, the aim of the present study was (i) to isolate bacteriophages against *B. cereus* group strains from different sewage pool samples (P1–P4); (ii) to characterize bacteriophages by the analysis of their host range, morphology, protein patterns and restriction enzyme digestion profiles; (iii) to determine the potential of isolated bacteriophages for the prevention and eradication *B. cereus* biofilm on stainless steel coupons.

## 2. Results

### 2.1. Isolation and Host Spectrum Evaluation of Bacteriophages

In total, 62 bacteriophages infecting the *B. cereus* group were isolated by enrichment from four different sewage pools (P1–P4). As mentioned in Table 1, the enrichment of the four pools (P1, P2, P3 and P4) allowed amplification of 16, 15, 19 and 12 bacteriophages infecting the *B. cereus* group, respectively. The purified bacteriophages were screened on the basis of host range analysis using the 174 *B. cereus* group strains in the collection. Plaques formed by phages on appropriate host bacteria were classified as clear, moderately clear and turbid (data not shown). Based on the quality of plaques and the lytic activity spectrum, seven phages (Ø1BC478; Ø1BC3990; Ø2BC4663; Ø3BC4663; Ø4BC4663(1); Ø4BC4663(2) and Ø1BC4663) were selected as having the broadest lytic spectrum (Appendix A). Plaque morphology on agar plates of isolated phages is illustrated in Figure 1. 

The isolated phages showed a lytic efficiency on tested *B. cereus* strains that ranged from 8.6% to 13.2% (Table 2).

### 2.2. Bacteriophage Characterization

#### 2.2.1. Morphological Observation 

Selected bacteriophages were examined by transmission electron microscopy (Figure 2). On the basis of their morphological features, all phages were considered to be part of the *Siphoviridae* family in Caudovirales [31], because they exhibited a typical icosahedral head and long, flexible and non-contractile tail structure in TEM micrographs. The dimensions and features of the phages are given in the Table 3. Within our phage samples, there were light and dark phage heads. Dark heads demosntrated loss of their nucleic acid [23,32].

#### 2.2.2. Protein Pattern Analysis

SDS-PAGE was used to determine the structural protein content of each phage (Figure 3). A similar profile was observed for phages Ø2BC4663 and Ø4BC4663(2), with four predominant protein bands corresponding to approximately 220, 180, 39 and 28 kDa. Phages Ø1BC3990, Ø3BC4663, Ø4BC4663(1) and Ø1BC4663 were found to produce two major protein bands corresponding, respectively, to 39, 32 kDa (Ø1BC3990); 39, 28 kDa (Ø3BC4663); 39, 31 kDa (Ø4BC4663(1)); and 42, 31 kDa (Ø1BC4663). Phage Ø1BC478 contained four major protein bands at molecular masses of approximately 120, 50, 39 and 28 kDa.

#### 2.2.3. Restriction Enzyme Digestion Pattern Analysis

The extracted isolated bacteriophages DNAs were digested with five restriction endonucleases (BamHI, EcoRI, HindIII, Hin6I and TaqI). Neither of the extracted phages DNAs were susceptible to BamHI and HindIII, whereas the EcoRI, Hin6I and TaqI enzymes resulted in very similar restriction profiles for selected phages (Figure 4). These results indicate that the selected phages have similar genome sizes and possibly are highly related. Selected phages have estimated genome sizes of around 37 kb, as calculated by adding together the fragment sizes obtained with the three enzymes above.

### 2.3. Bacteriophage Action on B. cereus Biofilms

#### 2.3.1. Effectiveness of Phage Treatment in Reducing Pre-formed Biofilms 

The eradication activity of each phage on pre-formed biofilms of 24, 48 and 72 h of the corresponding *B. cereus* host strain was evaluated by stainless steel application. Phage post-treatment of different preformed biofilms was performed using appropriate phage dilutions (10^5^–10^8^ PFU/mL), followed by six hours of incubation at 30 °C. The phage post-treatment effects on preformed biofilms are presented as percentage reduction of the biofilm in Figure 5.

All phage titrations (10^5^–10^8^ PFU/mL) were found to have removal efficiency against 24 h-old biofilm. In fact, bacteriophage post-treatments of each tested bacteriophage with titers of 10^8^, 10^7^, 10^6^ and 10^5^ PFU/mL reduced the corresponding pre-formed 24 h-old biofilm by 28.0 to 59.4%, 22.5 to 49.1%, 18.4% to 42.1% and 6.1 to 32.2%, respectively. The highest biomass removal activity of 59.4% was observed with Ø4BC4663(2) post-treatment at a titer of 10^8^ PFU/mL against 24 h-old biofilm (Figure 5F). Results showed that only higher phage titrations were found to be effective on already formed 48 h-old biofilm (≥10^6^ PFU/mL). In the 48 h-old biofilms, the highest removal efficiency of approximatively 37% was observed with Ø1BC3990 phage post-treatment at a titer of 10^7^ and 10^8^ PFU/mL (Figure 5B), whereas no eradication activity against 48 h-old biofilm was found with Ø2BC4663 (Figure 5C), Ø4BC4663(1) (Figure 5E) and Ø4BC4663(2) (Figure 5F) post-treatment at a titer of 10^5^ PFU/mL. Removal efficiency of phage post-treatment was dramatically decreased for the 72 h-old biofilm samples. No eradication against 72 h-old biofilms was detected for all phages titrations of both phage Ø1BC3990 (Figure 5B) and Ø2BC4663 (Figure 5C). However, the Ø1BC4663 post-treatment with titration of 10^8^ PFU/mL was found to be the most effective against 72 h-old biofilms, resulting in biomass removal of 25.6% (Figure 5G).

#### 2.3.2. Effectiveness of Phage Treatment in Inhibiting Biofilms 

The results of bacteriophage treatment of each isolated phage with different titers (10^5^, 10^6^, 10^7^ and 10^8^ PFU/mL) in inhibiting biofilm formation at different incubation periods (24, 48 and 72 h), are shown in Figure 6. All tested phage titrations prevented biofilm formation at the end of the 24 h and 48 h incubation periods, by 65.3 to 97.1% and 48.2 to 91.9%, respectively. Due to the long incubation period (72 h), the inhibition capability of all tested bacteriophages was found to be less effective at the end of the 72h incubation period. The log reduction percentage ranged from 16.8 to 28.4%, 37.9 to 58.3%, 66.7 to 76.8% and 79.1 to 87.1% for 10^5^, 10^6^, 10^7^ and 10^8^ PFU/mL, respectively. The highest inhibition, of 97.1%, was observed with Ø4BC4663(1) treatment at a titer of 10^8^ PFU/mL against the host strain biofilm formation after 24 h (Figure 6E). For phage treatment against biofilm formation after 48h of incubation, the highest inhibition, of 91.1%, was observed with phage Ø3BC4663 treatment at a titer of 10^8^ PFU/mL (Figure 6D). The lowest inhibition efficiency was detected for Ø1BC478 treatment (10^5^–10^8^ PFU/mL) against biofilm formation after different incubation periods (24, 48 and 72 h) (Figure 6A).

## 3. Discussion

In this study, we first studied the prevalence of *B. cereus* phages in sewage pool samples. A specific enrichment phase using a mix of 174 *B. cereus* strains was developed to allow the amplification of phages present in very minute numbers in a sample and, thus, to enhance the probability of isolating a wider range of phages quickly and at low cost. Previous studies [23,25] have used such an experimental protocol to isolate a variety of phages infecting *Campylobacter jejuni* and *Vibrio harveyi* strains. Among the 62 isolated phages, seven phages were chosen for further characterization, due to their host spectrum of activity and plaques quality (Appendix A). The lytic efficiency of the selected phages on tested *B. cereus* strains varied from 8.6 to 13.2% (Table 2). Although the majority of bacteriophage studies have concentrated on the characterization of high percentage lysis bacteriophages and their application as agents for biocontrol in the food industry, our study aims to test the possibility of using low percentage lysis bacteriophages in biofilm treatment. Indeed, and as reported in previous work [33], low percentage lysis bacteriophages can more effectively lyse a small number of host strains. Therefore, a broadening of the host spectrum of a bacteriophage is often counterbalanced by a limitation of the virulence of that bacteriophage towards its own host. [34,35]. The application of bacteriophages that act only on a specific host can simultaneously reduce the likelihood of developing resistance at the community level and have a relatively low impact on the microbial flora [36]. Furthermore, these bacteriophages can be combined to be exploited as novel biocontrol agents in a cocktail form [37,38] or explored for specific detection.

TEM analysis showed that all isolated *B. cereus* bacteriophages are members of the *Siphoviridae* familiy. This finding is consistent with previous studies, which identify the morphology of *B. cereus* bacteriophage isolates as part of the *Siphoviridae* family [28,39]. In contrast, other studies have showed that bacteriophages infecting the *B. cereus* group belonged to the *Myoviridae* family [27,29,32,40] as well as to the *Tectiviridae* family [41]. 

The structural proteins of the seven phages isolated were analyzed by using SDS-PAGE. The major structural proteins and several minor proteins were identified in the isolated phages. In order to evaluate the protein molecular weights in the samples, the migration of each band was checked against the migration patterns of standard proteins with known molecular weight. The *B. cereus* phages Ø2BC4663 and Ø4BC4663(2) showed a similar banding pattern. However, the protein profile varied among the remaining phages. Hence, the structural proteins are unique to each phage and depend on their morphotype, even though the phage is part of the same family. Similar results have been reported previously using *Escherichia coli* [42] and *Salmonella* bacteriophages [43]. The difference in protein profiles could explain phage host and plaque morphology diversity. Therefore, structural protein analysis was suggested as a method of determining phage morphotype, characterization and differentiation [44].

According to the results of this study, all of the selected phages have the same restriction pattern, with an estimated genome size of around 37 kb. These results may suggest that the seven selected phages are quite similar in genome size and may also have similar DNA sequences. The size of the phages’ genomes is comparable to that published in previous studies [45,46]. However, phages specific to the *B. cereus* bacteria have been described as having a genome size of more than 130 kb [32,47,48].

Comparative analysis of restriction fragment profiles showed that the genomes of the isolated phages were very similar. These results are consistent with those obtained in previous studies, which reported that several phages specific to the *B. cereus* group exhibited high genetic congruence [46,49,50,51,52]. The results of these studies suggest that bacteriophages with similar genomes are likely to be found in a variety of environments. Although the genomes of these phages are generally very similar, several genes show a great diversity, and very variable regions in the genomes have been observed, such as the genes coding for proteins having functions in lysis of the host and in DNA replication [46]. Sequencing the genome of the phages isolated in this work could allow a better comparison.

The present study evaluated whether isolated bacteriophages could prevent or eradicate *B. cereus* biofilm formation on a stainless steel surface, commonly found in the food industry. It was found that phage treatment at all titers (10^5^–10^8^ PFU/mL) was more effective in reducing biofilm formation (ranging from 0.1 to 1.5 log_10_) than eradicating already formed biofilms (from 0 to 0.5 log_10_). The low eradication percentage reported for 48 h- and 72 h-old biofilms could be attributed to the complex structure of preformed biofilms, that can pose a serious obstacle for phage penetration in multiple ways, such as the diffusion barrier imposed by the biofilm matrix and the physiological and metabolic state of biofilm cells [17]. 

*B. cereus* biofilms are composed mainly of vegetative cells, but during the maturation and aging process, *B. cereus* has the ability to form spores among established biofilms. Another study showed high levels (over 50%) of sporulation in 48 h submerged biofilms of different strains of the *B. cereus* group on stainless steel slides [53]. Sporulation of *B. cereus* in biofilms has been evidenced in several studies, although further studies are needed to determine whether biofilms play a role in sporulation and the direct link between biofilm formation and *B. cereus* sporulation.

The production and accumulation of enterotoxins/emetic toxins in the biofilm matrix may function as an autoinducer of quorum sensing and, in particular, may contribute to biofilm development and play an anti-competitive or defensive role in protecting an inhabited niche by preventing invasion by other species [54].

During the initial step of attachment in the biofilm formation process, bacterial cells may detach from the surface and return to a planktonic phase to colonize a new surface and, thus, bacteriophage treatment has high efficiency against these planktonic bacteria, as demonstrated in previous studies [55,56]. Therefore, the application of bacteriophages during the bacterial attachment step is more effective than on mature biofilms. Similar findings were reported during the application of bacteriophage P22 to reduce attachment and biofilms of *Salmonella typhimurium* on a stainless steel surface [18]. The researchers found that treatment with phage P22 was significantly more effective in inhibiting biofilm formation (from 0.08 to 1.2 log_10_) than in eradicating pre-formed mature biofilm, after treatment (from 0 to 0.5 log_10_). Likewise, a study on the application of bacteriophages to reduce the attachment and biofilms of *E. coli O157:H7* on a stainless steel surface [57] showed a reduction of 1.2 log_10_ CFU/coupon after bacteriophage treatment of bacterial attachment; however, the same bacteriophage treatment was not able to significantly reduce pre-formed biofilms under the same conditions.

Our results show that the duration of phage treatment, as well as the age of the biofilm, influences the efficiency of inhibition of biofilm formation and eradication of mature biofilms of *B. cereus*. This is in agreement with the results of other studies [10,57] that have revealed significant reductions in biofilms (ranging from 1 to 6 log10), dependent upon biofilm constituents, biofilm age, phage selection, and treatment duration.

As mentioned above, for mature pre-formed biofilm, the rate of biomass reduction was dependent on the biofilm matrix, that would render phage penetration difficult, rather than on the physiological status of the cells. Indeed, bacteriophages preferably infect planktonic cells in the exponential phase [58], such as the newly divided biofilm-surface bacteria and, thus, may not present sufficient efficiency against persistent cells with reduced metabolic activity, enclosed in biofilms, resulting in reducing bacteriophage treatment effectiveness [59,60].

For controlling biofilm formation, bacteriophage treatment is not effective enough when it is extended. Indeed, biofilm formation is concluded at the end of the incubation period, due to the decrease in phage concentration [18] or the appearance of resistance mechanisms of bacteria against phages, which negatively affects phage treatment efficacy [18,19,61]. As reported in previous studies [62,63], phage resistance within biofilms in laboratory conditions could be due to mutations in bacterial receptors or the loss of phage receptors. However, phages can overcome bacterial resistance by adapting to new receptors [64]. Moreover, during bacteriophage-mediated lysis, bacterial extracellular DNA can release and accumulate to a sufficient concentration, which would explain the increase in biofilm levels at this time [65].

## 4. Materials and Methods 

### 4.1. Bacterial Strains and Culture Conditions 

The bacterial strains used in this study for phage induction and host spectrum consisted of a collection of 174 *B. cereus* group strains, previously isolated from a total of 687 Tunisian food samples (cereals; spices; cooked food; canned products; seafood products; dairy products; fresh-cut vegetables; raw and cooked poultry meats) [66]. Brain heart infusion medium (BHI) (Fisher Bioblock, Illkirch, France), supplemented with yeast extract (YE) (Fisher Bioblock), was used to grow the *B. cereus* strains used in this study. All BHI-YE plates or broth cultures were incubated for 24 h at 30 °C. 

### 4.2. Isolation of B. cereus-Specific Phages and Host Spectrum Evaluation

#### 4.2.1. Sampling

Bacteriophages were isolated from sewage, before and after treatment, collected from a wastewater treatment plant (Rennes, France) and from a total of 50 samples containing animal wastewater (23), slurry (7), fecal waste (11) sewage (5) and soil samples (4) collected from an experimental farm specializing in feed production for pigs, poultry and rabbits (Paris, France). For the absolute presence of bacteriophages, all samples were divided into 4 groups (P1–P4). The first group (P1) was made up of 25 samples taken from the experimental farm. The pooled sample of the first group combined 15 animal wastewater samples (settling ponds (7), pig effluent (5) and poultry effluent samples (3)), 6 samples of fecal waste and 4 samples of soil, altogether. The second group (P2) contained the remaining samples (25) taken from the experimental farm. Seven slurry, 5 fecal waste, 5 settling pond, 5 sewage and 3 rabbit effluent samples made up the combined sample of P2. For the third (P3) and fourth (P4) groups, each contained a mixture of sewage effluent samples, taken from the wastewater treatment plant before and after treatment, respectively.

#### 4.2.2. Enrichment Phase 

The collection of 174 *B. cereus* group strains was used to multiply bacteriophages in each pool sample (P1–P4) as follows: after incubation of each *B. cereus* group strain in BHI-YE broth for 5 h at 30 °C, 500 μL of each culture was mixed and added to 500 mL of BHI-YE broth to form the host culture broth. After 3 h of contact, 10 mL of each pooled sample (P1–P4) were added separately to the host culture broth. After incubation at 30 °C for 24 h, the culture supernatant was harvested by centrifugation at 7000 rpm for 20 min and sterilized by filter-sterilization (0.22 µm in pore size) (Starstedt, Nümbrecht, Germany). The filtrates obtained by this process were stored at 4 °C.

#### 4.2.3. Phage Isolation, Propagating and Purifying

The bacteriophages infecting the *B. cereus* group were isolated using a spot assay technique [23], with slight modifications. Briefly, 5 mL of BHI-YE soft agar (0.8% agar) inoculated with 100 µL of each *B. cereus* strain culture were overlaid on a pre-solidified BHI-YE agar (1.5% agar) plate. Once the overlay was dry, 10 µL of enriched suspension of each pooled sample (P1–P4) was spotted on the plate. The plates were left for 30 min until the spots were dry and then incubated overnight at 30 °C. In the case of phage detection (forming of lysis zones), plaques were individually picked up from the agar, dispensed into a sterile falcon tube containing 3 mL of BHI-YE broth and left at room temperature overnight to allow the phages to be diffused from the agar. Then, samples were mixed gently, and the liquid was filtered with a 0.22 μm filter. Subsequently, a phage titer was determined following a soft agar overlay method [29,32,67,68]. Briefly, the filtrate of the propagated plaque was diluted serially 10 times in TS buffer (8.5 g NaCl and 1 g tryptone per liter), mixed with 200 µL of each cultured *B. cereus* host strain and left for 15 min at room temperature to allow phages to adhere on the bacterial cells. Once the incubation was completed, the culture was added to 3 mL of soft agar (BHI-YE broth with 0.8% agar) and poured onto pre-solidified BHI-YE agar (1.5% agar). The plate was then incubated at 30 °C overnight. After this, plates were then observed for plaque formation, and this was expressed as plaque forming units/mL (PFU/mL) of supernatant. A sample was considered positive for phages when plaques were noted on the bacterial lawn of the plates. This procedure was repeated at least three times until the morphologies of the plaques became consistent. Phage-free cultures (containing only bacterial host) and host-free cultures (containing only phage) were used as controls. 

#### 4.2.4. Host Range Assay

A total of 62 phages infecting the *B. cereus* group were isolated. Host range analysis was performed to separate the broad-spectrum lytic phages from the 62 generated phage lysates. The host range of the 62 phages was tested by a spotting technique [32] on 174 *B. cereus* group strains, conducted by spotting 10 μL of purified phage lysate onto host bacterial lawns prepared on a pre-solidified BHI-YE agar (1.5% agar) plate. The plates were left for 30 min until the spots were dry and then incubated overnight at 30 °C for lysis zone formation. As a control, each bacterial strain was also infected using sterile phage buffer. Lytic effectiveness of each phage lysate on the host bacteria was confirmed by plaque formation. Plaques were classified into the following three categories: clear, moderately clear, and turbid (data not shown). Seven *B. cereus* phages were selected for having produced clear plaques against a maximum number of bacterial strains (Appendix A) and were subjected to further characterization steps.

### 4.3. Bacteriophage Characterization 

#### 4.3.1. High-Titer Phage Stock Preparation 

The phage concentration step was performed according to a modified method described in previous studies [32,67]. The filtrate of purified phages was diluted serially by ten-fold in TS buffer to achieve a concentration allowing confluent lysis of the host in a soft agar overlay plate. The dilution producing the most confluent lysis was chosen. Each phage stock was then plated by using the overlay method to achieve confluent lysis on 20 plates for each sample. After an overnight incubation, 5 mL of BHI-YE broth was added to each of these 20 plates. The suspension was recovered and left at 4 °C for 3 to 4 h, to allow the phages to diffuse from the overlay. The suspension was then centrifuged at 9000 rpm for 20 min, and this was sufficient to separate the overlay agar. The supernatant was harvested and filtered (0.22 μm), followed by ultracentrifugation at 35,000 rpm (Sorvall Discovery 90SE ultracentrifuge; T-865 rotor) for 2 h at 20 °C. The supernatant was then gently discarded, and the pellet was resuspended in 300 µL of TE buffer (10 mM Tris-HCL, 1 mM EDTA; pH = 8) that was added to each tube. The titer of the phage stock was then counted by the overlay method, using the appropriate host strain.

#### 4.3.2. Morphology Observation by Transmission Electron Microscopy

Using a transmission electron microscope (TEM), phage morphologies were examined by a single negative staining method. The target titer of the stocks was >10^10^ PFU/mL to allow visualization by electron microscopy. Thirty microliters of phage stock was spotted onto 200-mesh copper grids with carbon-coat formvar films (Electron Microscopy Sciences, Hatfield, PA, USA) and left for 1 min. The grids were then stained with 2% aqueous uranyl acetate for 1 min. Excess liquid was blotted with filter paper and the grid was air dried. Bacteriophage morphology was examined by electron microscopy in a JOEL JEM-1400 transmission electron microscope.

#### 4.3.3. Structural Protein Analysis 

As described by Geng et al. (2017) [39], phage stock (10^9^ PFU/mL) was diluted 1:1 in a dissociation buffer containing 95% (*v/v*) of 2× Laemmli Sample Buffer (Bio-Rad, Marnes-la-Coquette France) and 5% (*v/v*) of ß-mercaptoethanol (Sigma-Aldrich, Saint Quentin, Fallavier, France) and then boiled for 10 min to denature the viral proteins. The phage proteins were separated by SDS-PAGE in precast 4–20% SDS-polyacrylamide gel (Mini-PROTEAN TGX ^TM^ ) (Bio-Rad). Electrophoresis was carried out at 200 V for approximatively 30 to 40 min. Gels were stained with Bio-Safe Coomassie blue G250 (Bio-Rad) for 1 to 2 h. Protein mass determination was performed using Precision Plus Protein^TM^ unstained standards (Bio-Rad).

#### 4.3.4. Restriction Digest Profile Analysis of *B. cereus* Bacteriophages

Genomic DNA from a high titer of phages, prepared as described above (Section 4.3.1), was extracted using the phenol/chloroform method according to Sambrook and Russell (2001) [69]. The phage suspension (1 mL) was treated with DNase (invitrogen, Thermo Fisher Scientific, Vilnius, Lithuania) for 30 min, followed by the addition of 40 μL of proteinase K (Qiagen, Hilden, Germany) and incubation at 56 °C for 30 min. Then, 100 μL of 10% SDS was added, and the solution was incubated at 65 °C for 30 min. To remove proteins from the phage suspension, the mixture was treated with equal volumes of phenol/chloroform/isoamyl alcohol (25:24:1, *v/v*) twice. Subsequently, phage DNA was precipitated by adding 1/10 volume of aqueous 3 mol sodium acetate (pH 4.8) and 2.5 volumes of 100% ethanol and washed with 70% ethanol, and finally dissolved in 40−50 μL of milliQ water (Sigma-Aldrich). The quality of the extracted DNA was evaluated by using a NanoDrop ND-1000 spectrophotometer (Nanodrop Technologies, Wilmington, NC, USA), as measured by the absorbance ratio at 260 and 280 nm. The quality of the extracted DNA was also examined by 1% agarose-gel electrophoresis at 80 V for about 90 min. The genomic DNA was used for a restriction enzyme treatment with BamHI, EcoRI, HindIII, Hin6I and TaqI in an appropriate buffer, according to the manufacturer’s instructions. The digested products were then analyzed by electrophoresis at 80 V for approximately 2 h on a 1% agarose gel in 1× TAE buffer (Promega, Madison, WI, USA) (Promega, Madison, WI, USA), and stained with ethidium bromide using a 10 kb SmartLadder (Eurogentec, Seraing, Belgium) as a molecular weight marker.

### 4.4. Assessment of Bacteriophage Action on B. cereus Biofilms 

#### 4.4.1. Preparation of Stainless Steel Coupons 

As stainless steel is frequently used in food processing areas, 304 L stainless steel coupons measuring 2 mm thick and 15 mm in diameter (Bretagne Laser, Guer, France), were used in this study. The coupons were prepared for treatment assays as previously described by Jan et al. (2011) [70]. After immersion in 2% (*v:v*) RBS 35 (Chemical Products R. BORGHGRAEF, Brussels, Belgium) in distilled water, the coupons were rinsed 5 times (for 5 min per rinse) with distilled water at 50 °C and 5 times with distilled water at room temperature. The coupons were then autoclaved for 20 min at 120 °C and transferred to the bottom of the wells of a 24-well microplate, using one coupon per well. 

#### 4.4.2. Reducing Pre-Formed Biofilm Experiment (Phage Post-Treatment)

An aliquot of 300 µL of each bacterial host used for propagation (strain 478, strain 3990 and strain 4663) [66] was taken from exponentially growing cultures (10^8^ CFU/mL) and deposited in each well. The plates were incubated statically for 24, 48 and 72 h at 30 °C. Following the incubation period, the cell suspensions were removed, and the coupons held in the wells were rinsed with 9 g/L NaCl solution in order to discard non-adherent cells. To assess the effectiveness of phage treatment on the reduction of preformed biofilm of the *B. cereus* group, the methodology described in previous studies [16,18] was followed with some adjustments. In brief, already formed biofilm samples on stainless steel coupons were treated with 300 µL of the appropriate phage dilutions (10^5^–10^8^ PFU/mL) and incubated for 6h at 30 °C. Tests were performed twice in triplicate. One triplicate included, and the other excluded, the addition of bacteriophages, with each sample being used as its own positive control. After the bacteriophage treatment, each well was rinsed with 9 g/L NaCl solution 3 times and allowed to air-dry. Then, each coupon was moved to a new tube filled with 5 mL of 0.02% aqueous Tween80 solution (Thermo Fisher Scientific) and five sterile glass beads (3 mm of diameter). The coupons were then vortexed for 1 min at the highest intensity. To separate residual phages, the recovered suspension was centrifuged for 10 min at 7000 rpm at 25 °C. Cell pellets were resuspended in 1 mL of TS buffer and enumeration of bacteria was carried out by a plate counting micro-method [71]. The CFU (colony forming unit) count per square centimeter was converted to a logarithmic value. Results are presented as the effectiveness of bacteriophage treatment on reducing pre-formed biofilm of the *B. cereus* group. The percentage of biofilm eradication was determined using the following formula [18]: [PK = (1 − 10^−LR^) × 100%] where PK is the percentage killing and LR is the log reduction [log_10_ (untreated viable cell density) −log_10_ (treated viable cell density)].

#### 4.4.3. Inhibiting Biofilm Formation Experiment (Phage Treatment) 

The protocol for evaluation of the inhibition potential of bacteriophages on *B. cereus* biofilm formation was adapted and optimized (with modifications) from previous work [18]. Briefly, each *B. cereus* host strain was cultured with appropriate phage suspensions adjusted to 10^4^–10^8^ PFU/mL in a test medium. The plates with stainless steel coupons were prepared as described in Section 4.4.1. An aliquot of 300 µL of each bacterial host (10^8^ CFU/mL) and 300 µL of the appropriate phage suspension, adjusted to 10^5^–10^8^ PFU/mL in TE buffer, were deposited in each well. The control well was inoculated only with host bacterial suspension. Tests were done in triplicate. The plates were incubated for 24, 48 and 72 h at 30 °C. Experimental data were obtained as described in Section 4.4.2. Results are presented as the effectiveness of bacteriophage treatment on the inhibition of *B. cereus* biofilm formation. The logarithmic reduction of biofilm cells was also calculated by the same formula given above.

## 5. Conclusions

The isolation and characterization of phages specific to the *B. cereus* group represents the first accomplishment of this study. Following characterization, the seven selected phages were found to be different in terms of identity. The second accomplishment of this work was to evaluate the effectiveness of phages to treat *B. cereus* attachment and biofilms on stainless steel surfaces. We found that phage treatment had the potential to inhibit up to 97.1% of biofilm formation and to reduce up to 59.4% of pre-formed biofilms. Our findings suggest that the use of bacteriophages in food processing facilities may reduce the possibility that finished products will be re-contaminated with *B. cereus*. Nevertheless, to evaluate the efficacity on real environment biofilms, some further experiments are needed, such as optimizing the bacteriophage cocktail to remove mixed *B. cereus* biofilm that occurs with other microflora on food contact surfaces. Moreover, the application of phage treatment as a highly potent approach to inactivate *B. cereus* attachment and biofilms in the food industry should be verified in a broader range of temperature, humidity and pH conditions.

## Figures and Tables

**Figure 1 pathogens-11-00872-f001:**
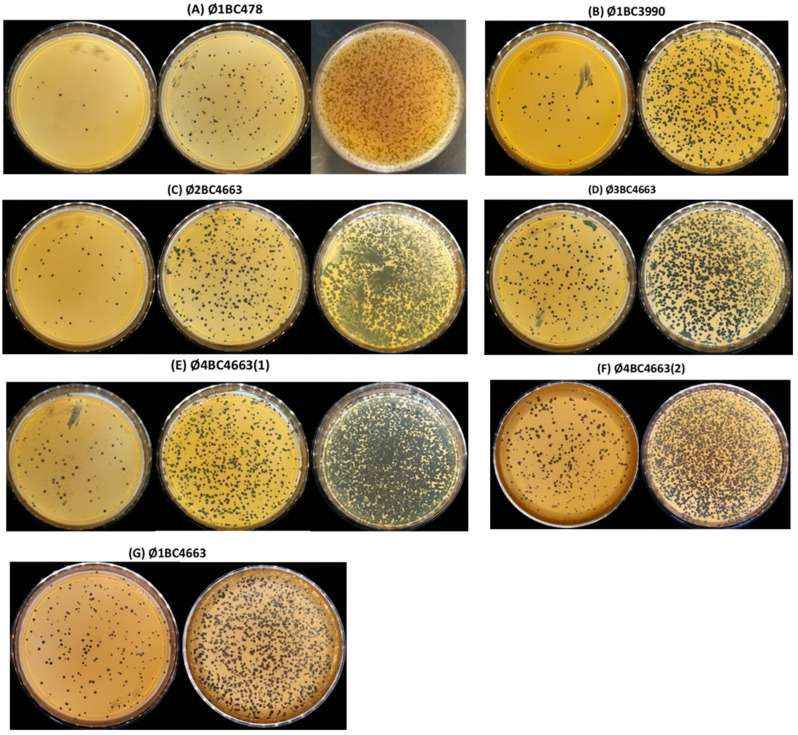
(**A–G**) Plaque morphology on agar plates of isolated phages. (**A**) Ø1BC478, (**B**) Ø1BC3990, (**C**) Ø2BC4663, (**D**) Ø3BC4663; (**E**) Ø4BC4663(1); (**F**) Ø4BC4663(2); and (**G**) Ø1BC4663.

**Figure 2 pathogens-11-00872-f002:**
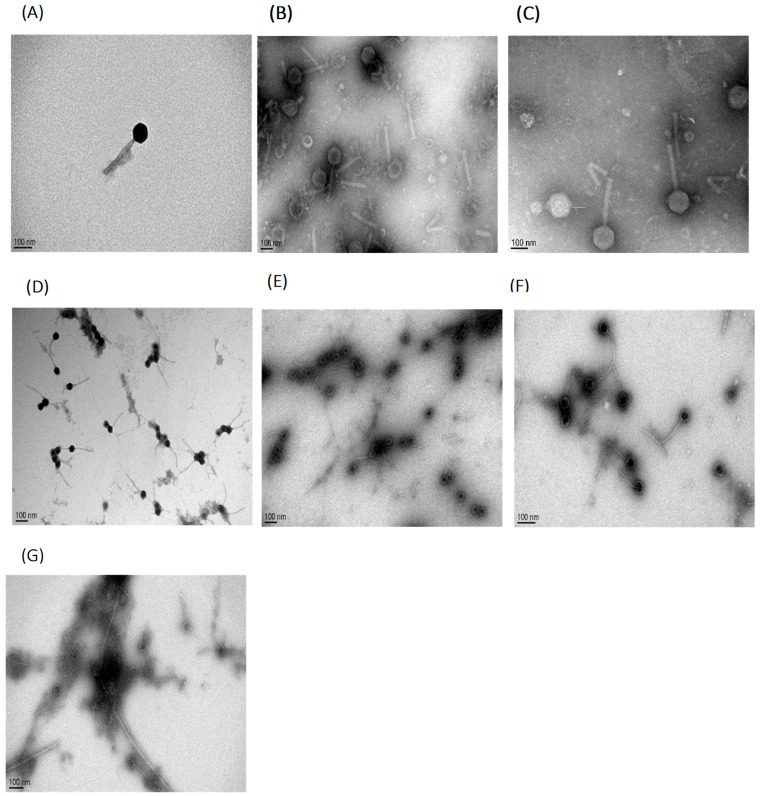
(**A**–**G**) Transmission electron micrographs of seven *B. cereus*- specific bacteriophages isolated in this study. (**A**) Ø1BC478; (**B**) Ø1BC3990; (**C**) Ø2BC4663; (**D**) Ø3BC4663; (**E**) Ø4BC4663(1); (**F**) Ø4BC4663(2); and (**G**) Ø1BC4663. The virions were negatively stained with uranyl acetate and visualized at a magnification of 100,000× (**B**,**D**,**F**), 120,000× (**C**,**G**) and 150,000× (**A**,**E**). The scale bar indicates 100 nm.

**Figure 3 pathogens-11-00872-f003:**
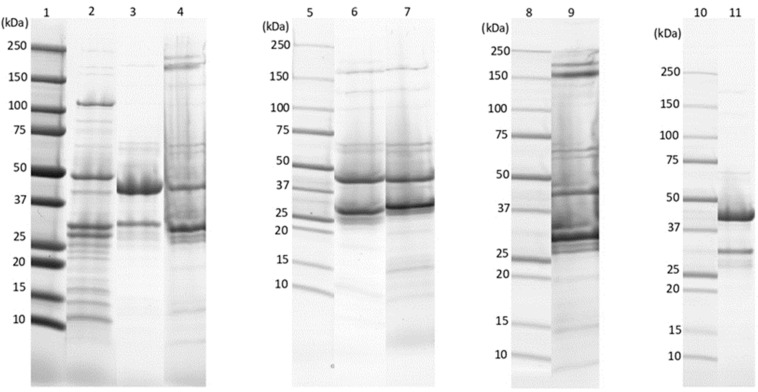
SDS-PAGE analysis of phage structural proteins. Lane 1; 5; 8 and 10: Precision Plus Protein^TM^ unstained standards (Biorads). Lane 2; 3; 4; 6; 7; 9 and 11: Ø1BC478; Ø1BC3990; Ø2BC4663; Ø3BC4663; Ø4BC4663(1); Ø4BC4663(2) and Ø1BC4663; respectively.

**Figure 4 pathogens-11-00872-f004:**
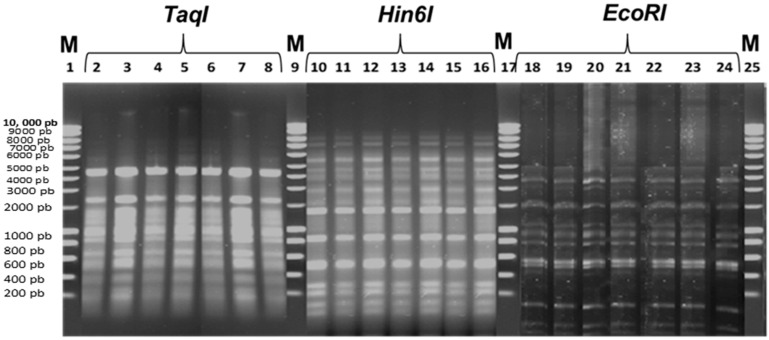
Restriction enzyme digestion pattern analysis of isolated bacteriophages DNAs. Lane 2 to Lane 8: TaqI digestion of Ø1BC478; Ø1BC3990; Ø2BC4663; Ø3BC4663; Ø4BC4663(1); Ø4BC4663(2) and Ø1BC4663, respectively. Lane 10 to Lane 16: Hin6I digestion of Ø1BC478; Ø1BC3990; Ø2BC4663; Ø3BC4663; Ø4BC4663(1), Ø4BC4663(2) and Ø1BC4663, respectively. Lane 18 to Lane 24: EcoRI digestion of Ø1BC478; Ø1BC3990; Ø2BC4663; Ø3BC4663; Ø4BC4663(1); Ø4BC4663(2) and Ø1BC4663, respectively. Lanes 1, 9, 17 and 25: Molecular marker (SmartLadder 10 kb).

**Figure 5 pathogens-11-00872-f005:**
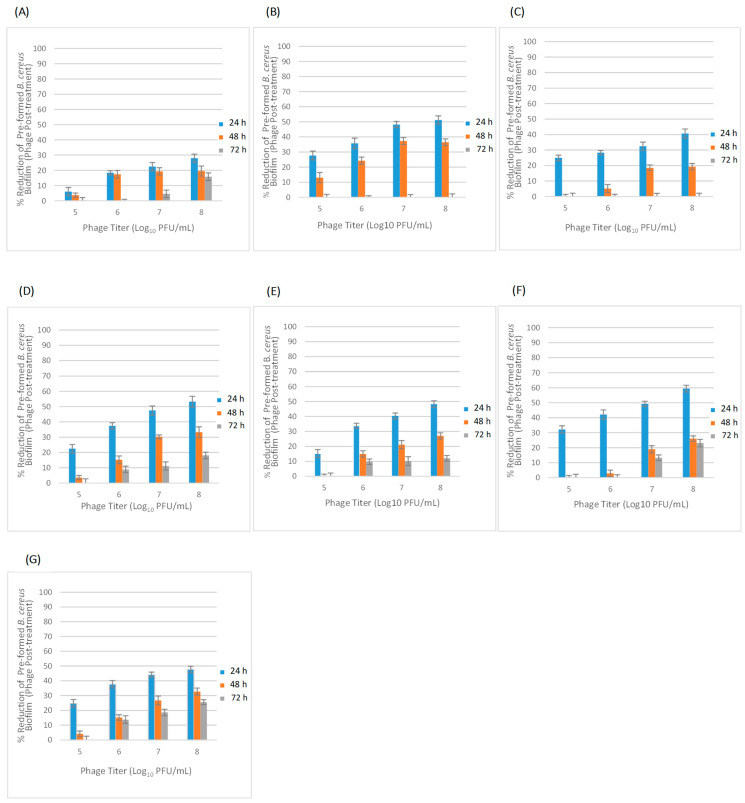
(**A**–**G**) Effect of phage post-treatments at different titrations (10^5^–10^8^ PFU/mL) on pre-formed biofilms of 24, 48 and 72 h and the reduction percentages of biofilm in a stainless steel application (bars represent standard deviation). (**A**) Ø1BC478; (**B**) Ø1BC3990; (**C**) Ø2BC4663; (**D**) Ø3BC4663; (**E**) Ø4BC4663(1); (**F**) Ø4BC4663(2); and (**G**) Ø1BC4663.

**Figure 6 pathogens-11-00872-f006:**
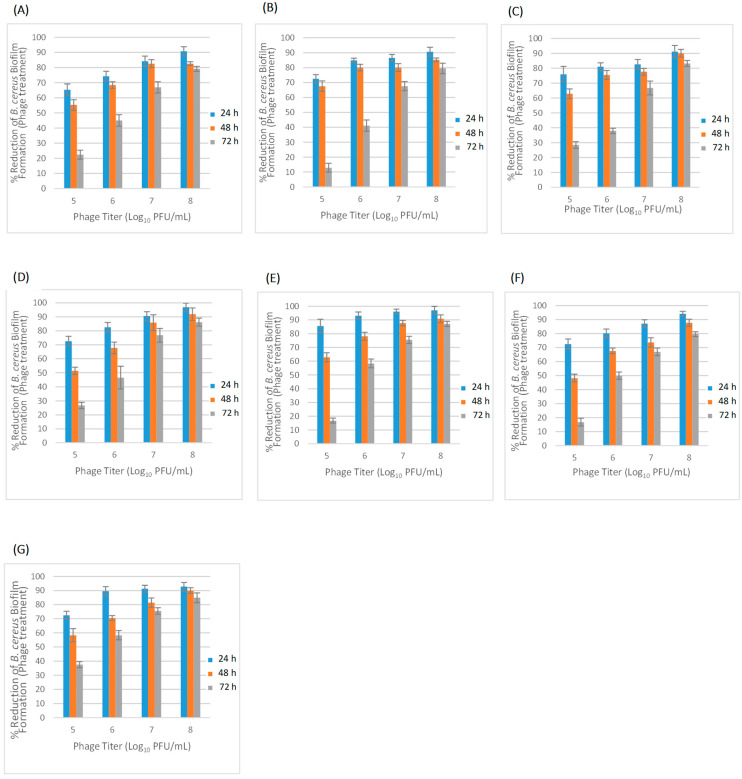
(**A**–**G**): Effect of phage treatments at different titrations (10^5^–10^8^ PFU/mL) on biofilm formation during incubation periods of 24, 48 and 72 h and the inhibition percentages of biofilm formation in a stainless steel application (bars represent standard deviation). (**A**) Ø1BC478; (**B**) Ø1BC3990; (**C**) Ø2BC4663; (**D**) Ø3BC4663; (**E**) Ø4BC4663(1); (**F**) Ø4BC4663(2); and (**G**) Ø1BC4663.

**Table 1 pathogens-11-00872-t001:** The details of isolation of *B. cereus* phages.

Pooled Sample	Number of Isolated Phages	Isolated Phages
P1	16	Ø1BC13; Ø1BC14; Ø1BC18; Ø1BC27; Ø1BC28; Ø1BC396; Ø1BC478; Ø1BC499; Ø1BC2893; Ø1BC2976; Ø1BC3296; Ø1BC3940; Ø1BC3990; Ø1BC4663; Ø1BC4785; and Ø1BC4851
P2	15	Ø2BC12; Ø2BC13; Ø2BC14; Ø2BC27; Ø2BC28; Ø2BC396; Ø2BC2893; Ø2BC2976; Ø2BC3296; Ø2BC3990; Ø2BC4663; Ø2BC4785; Ø2BC4851; Ø2BC4855; and Ø2BC5659
P3	19	Ø3BC13; Ø3BC14; Ø3BC18; Ø3BC24; Ø3BC27; Ø3BC396; Ø3BC478; Ø3BC2893; Ø3BC3296; Ø3BC3940; Ø3BC3990; Ø3BC4160; Ø3BC4165; Ø3BC4663; Ø3BC4785; Ø3BC4851; Ø3BC4855; Ø3BC5659; and Ø3BCE10
P4	12	Ø4BC13; Ø4BC24; Ø4BC27; Ø4BC396; Ø4BC478; Ø4BC2893; Ø4BC3990; Ø4BC4663(1); Ø4BC4663(2); Ø4BC4785; Ø4BC4851; and Ø4BCE10

**Table 2 pathogens-11-00872-t002:** Percentage infectivity of each isolated bacteriophage, using a spot test method against 174 *B. cereus* strains.

Isolated Phages	Percentage Lysis (%)
Ø1BC3990	13.2
Ø1BC478	13.2
Ø3BC4663	12.6
Ø2BC4663	10.3
Ø4BC4663(2)	10.3
Ø4BC4663(1)	9.7
Ø1BC4663	8.6

**Table 3 pathogens-11-00872-t003:** Morphological specificity of *B. cereus* phages as demonstrated by transmission electron microscopy.

Phages	Characteristics	Total Length (±Standard Deviation) (nm)	Head Diameter (±Standard Deviation) (nm)	Family
Ø1BC478	icosahedral head, non-contarctile tail	330.5 ± 90.2	75.0 ± 31.8	*Siphoviridae*
Ø1BC3990		411.4 ± 86.9	125.5 ± 20.8	
Ø2BC4663		340.0 ± 93.8	113.3 ± 38.9	
Ø3BC4663		355.6 ± 59.6	86.4 ± 27.6	
Ø4BC4663(1)		200.3 ± 74.9	87.5 ± 40.3	
Ø4BC4663(2)		337.7 ± 60.8	112.0 ± 29.9	
Ø1BC4663		314.2 ± 87.9	97.61 ± 18.9	

## Data Availability

Not applicable.

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
