# Peer review of "Isolation, Partial Characterization and Application of Bacteriophages in Eradicating Biofilm Formation by Bacillus cereus on Stainless Steel Surfaces in Food Processing Facilities"

_pathogens, 2022, doi:10.3390/pathogens11080872_

Round 1
Reviewer 1 Report
In this paper, the authors give a preliminary characterization of seven bacteriophages. These phages show the highest lytic properties from a collection of 68 phage isolates obtained from 4 pulls of 50 samples. These seven isolates were tested for specificity on a collection of 174 isolates of the B. cereus group isolated from Tunisian foods. For phage isolates, virion morphology was shown by TEM, capsid protein composition by sdsPAGE, restriction mapping of genomic DNA, the ability to eradicate biofilms and prevent biofilm growth. Phages were purified using CsCl ultracentrifugation. The genomic DNA of these seven phage isolates has similar restriction profiles, but the sdsPAGE profiles of virion proteins of some phages, as well as the effect on biofilms, differ.
Major comments
This report could have an important impact on the study of the potential application of bacteriophages in the Tunisian food industry, as it is based on local bacterial isolates. However, the article lacks important information, and the quality of some experiments is unsatisfactory. Significant revision of the manuscript is required:
(1) Please add an additional table to show the specificity of each of the seven phage isolates for the 174 bacterial strains of the Bacillus cereus group. This will allow comparison of phage range spectra and finding phage combinations to formulate phage cocktails to cover a large population of Bacillus cereus bacteria specific to Tunisia.
(2) Please add plaque morphology images for all 7 phage strains on host strains (used for propagation).
(3) Figure 1. The quality of transmission electron microscopy must be improved. From Fig. 1 E,F,G it is impossible to make any conclusions. The scale bar must be legible in all figures.
Minor comments:
Abstract – please indicate base of the logarithm near “log” like in materials and methods.
Fig 3. Please add sizes of DNA marker bands
Reviewer 2 Report
Comments and Suggestions for Authors
The paper presents some new aspects in the field of biological control of Bacillus cereus. This species of bacteria is considered as widespread foodborne pathogen. The first aim of the study was the isolation and characterization of infecting B. cereus bacteriophages. Subsequently, the authors examined the effectiveness of phages to treat B. cereus attachment and biofilms on stainless steel surfaces. The experiment was well planed. The Introduction is well written and provides information about the possibilities of using work results in the food industry. The aim of the paper has been clearly formulated. Minor revision for the English language are requested (some phrases in the text are incomprehensible). To attain publication quality I have the following suggestions:
-In the Results and Discussion section (especially in chapters 2.3.1 and 2.3.2.): the text should refer to specific Figures.
Example: The highest inhibition of 97.1% was observed with Ø 4BC4663 (1) treatment at a titer of 108 PFU / mL against 185 the host strain biofilm formation after the 24h (Figure 5E).
This will make it easier for the reader to interpret the presented data.
-I propose to improve the graphical quality of Figures.
-Line 173: 108 PFU/ml instead 108 PFU/ml
-Line 177: bacteriophage instead Bacteriophage
-Line 232: Therfore, structural protein instead Therfore, Structural protein
-Line 279 -283: Is this a description of another research example? This sentence is incomprehensible. I propose to write this sentence again.
- A double bibliography appears many times in the discussion (authors' names and numbers): Lines: 275, 285-286, 300.
-Line 335: What was the composition of this substrate?
-Lines 363-364: This sentence should be removed.
-Line 445: I did not find information about these strains in earlier text.
-Line 456: …. each sample being used as its own positive control. After bacteriophage treatment instead …. each sample being used as its own positive control After bacteriophage treatment
-References require editorial correction: no commas after surnames (Lines 512-513, 605, 615), too large spaces between lines (Lines 569-588), remove & (Line 613)

Round 2
Reviewer 1 Report
The article can be accepted, but I want to point out one very important issue regarding the confusion of the taxonomy of bacteriophages and their virion morphotypes. The taxonomy of viruses is currently based on genome sequences (more information can be found on the website of the International Committee on Taxonomy of Viruses). Please use the terms siphoviridae/mioviridae/tectaviridae not as taxonomic terms, but as bacteriophage virion mophotypes throughout the entire text of the article.